# Modification of Microstructure and Mechanical Properties of Extruded AZ91-0.4Ce Magnesium Alloy through Addition of Ca

**DOI:** 10.3390/ma17133359

**Published:** 2024-07-08

**Authors:** Fengtao Ni, Jian Peng, Xiangquan Liu, Pan Gao, Zhongkui Nie, Jie Hu, Dong Zhao

**Affiliations:** 1State Key Laboratory of Mechanical Transmissions, College of Materials Science and Engineering, Chongqing University, Chongqing 400044, China; nfengtao@163.com (F.N.); liuxiangquan2022@163.com (X.L.); niezhongkui@foxmail.com (Z.N.); hujie0676@163.com (J.H.); zhaodong@cqu.edu.cn (D.Z.); 2Jiaozuo Gaozhao Magnesium Alloy Co., Ltd., Jiaozuo 454950, China; gp385385@126.com

**Keywords:** AZ91 magnesium alloy, rare earth element Ce, Al_2_Ca phase

## Abstract

The effect of the addition of alkali earth element Ca on the microstructure and mechanical properties of extruded AZ91-0.4Ce-xCa (x = 0, 0.4, 0.8, 1.2 wt.%) alloys was studied by using scanning electron microscopy, transmission electron microscopy, and tensile tests. The results showed that the addition of Ca could significantly refine the second phase and grain size of the extruded AZ91-0.4Ce alloy. The refinement effect was most obvious when 0.8 wt.% of Ca was added, and the recrystallized grain size was 4.75 μm after extrusion. The addition of Ca resulted in the formation of a spherical Al_2_Ca phase, which effectively suppressed the precipitation of the β-Mg_17_Al_12_ phase, promoted dynamic recrystallization and grain refinement, impeded dislocation motion, and exerted a positive influence on the mechanical properties of the alloy. The yield strength (YS), ultimate tensile strength (UTS), and elongation (EL) of the AZ91-0.4Ce-0.8Ca alloy were 238.7 MPa, 338.3 MPa, and 10.8%, respectively.

## 1. Introduction

Magnesium (Mg) alloys are the lightest metal structural material in practice, and it is of great significance to achieve energy saving and emission reduction by using lightweight transportation vehicles [1,2,3,4]. As a structural part of a vehicle, its low absolute strength and poor plastic-forming ability at room temperature have become the resistance to expand its application [5,6,7]. The AZ91D magnesium alloy has good mechanical properties at room temperature, and its second phase is mainly α-Mg and β-Mg_17_Al_12_ [8,9]. The β-Mg_17_Al_12_ phase is a hard and brittle phase, which is unfavorable to the mechanical properties of the alloy. Therefore, we want to further improve the mechanical properties by using alloying modifications.

Owing to their high solid solubility in Mg, rare earth elements including cerium (Ce) have been widely used to improve the mechanical properties of Mg alloys [10,11]. Wang et al. [12] investigated the effect of Ce on extruded AZ80 alloys. The addition of Ce can refine the grain and the second phase, and the formability at room temperature is significantly improved. R. Krishnan et al. [13] found that the addition of Ce makes the β-Mg_17_Al_12_ phase in AZ91D alloys change from a continuous network to a small and dispersed distribution along the grain boundaries.

As an alkaline earth element, Ca has a large atomic size relative to rare earth elements, and its high solubility in the Mg matrix is about 0.34 at.% at 400 °C [5]. The addition of an appropriate amount of Ca to AZ series alloys has been demonstrated to enhance the mechanical properties of these alloys [14,15]. Zhang et al. [16] found that the addition of low Ca content greatly weakened the texture of Mg alloys, refined the grain size, and obtained a higher elongation. Huang et al. [17] found that in the extruded Mg-6Al-6Ca alloy, a large number of long rod-like Al_2_Ca phases were dispersed at the grain boundaries of the alloy, resulting in limited dynamic recrystallization (DRX), and the majority of the α-Mg matrix existed in the form of a non-DRX type containing a high density of low-angle grain boundaries (LAGBs).

However, the optimal addition of Ce and Ca in AZ91-Ce-Ca alloys has not been systematically reported. Therefore, four kinds of AZ91-0.4Ce-xCa (x = 0, 0.4, 0.8, 1.2 wt.%) alloys were designed to study the effects of different contents of Ca on the microstructure and mechanical properties of AZ91-0.4Ce, and to determine the optimal amount of Ca addition.

## 2. Materials and Methods

The cast raw materials of this experiment were pure Mg (99.99 wt.%), pure Al (99.95 wt.%), pure Zn (99.95 wt.%), Mg-30 wt.% Ce master alloy, and Mg-30 wt.% Ca master alloy. A resistance furnace with a rated power of 5 kW and a low-carbon steel crucible were used in the melting process. The Mg blocks were added into the preheated crucible, then heated to 750 °C. After the Mg blocks were completely melted, the pure Al, pure Zn, and master alloy with a preheating temperature higher than 150 °C were added successively. After all of the materials were melted, they were slagged and stirred, held for 30 min, cooled down, kept warm at 720 °C for 15 min, and then cast into a low-carbon steel mold at 350 °C. CO_2_ and SF_6_ shielding gases were always present throughout the smelting process. Finally, four samples of AZ91-0.4Ce-xCa (x = 0, 0.4, 0.8, 1.2 wt.%) and the scanning electron microscope (SEM) images of the as-cast AZ91-0.4Ce-xCa alloys are shown in Figure 1. The EDS spectrum results are shown in Appendix A.

After homogenization at 400 °C for 12 h, the ingot was turned into a Φ82 mm × 50 mm cylindrical ingot and extruded into a Φ16 mm bar on a 500 T horizontal extruder. After extrusion, the ingot was air-cooled to room temperature. The extrusion temperature was 370 °C, the extrusion speed was 1 m/min, and the extrusion ratio was 25:1.

The actual chemical composition of the as-cast AZ91-0.4Ce-xCa alloys was determined by using an XRF-1800CCDE (Shimadzu, Kyoto, Japan) X-ray fluorescence spectrometer, and the results are shown in Table 1. The alloy phase type was determined by using a D/MAX-2500PC (Rigaku, Kyoto, Japan) X-ray diffractometer (XRD) with a scanning speed of 4°/min and a scanning angle of 10–90°. A field-emission scanning electron microscope (SEM; JEOL-7800F), equipped with electron backscattered diffraction (EBSD), electron dispersive spectrometry (EDS), and back-scattered electron spectrometry (BSE), and transmission electron microscopy (TEM) were used to observe the microstructure. Among them, the SEM sample preparation process was as follows: the sample was polished to 1500# with metallographic sandpaper, and then corroded with saturated picric acid for 10–20 s. The TEM sample preparation process was as follows: the 0.5 mm thick sample underwent initial mechanical polishing to reduce its thickness to 50 μm, followed by subsequent low-temperature ion thinning until the process automatically halted upon perforation. The mechanical properties were obtained by stretching the CMT-5105 (SENS, Kyoto, Japan) electronic universal testing machine at a speed of 1 mm/min at room temperature, and the average value of 3 parallel experiments was taken as the final result.

## 3. Results

### 3.1. Microstructure of Extruded AZ91-0.4Ce-xCa Alloys

According to Figure 1, β-Mg_17_Al_12_ phases were distributed along the grain boundaries in discontinuous mesh or strip structures without Ca addition. The β-Mg_17_Al_12_ phases were obviously refined after adding Ca; and the content of the second phase increased and the distribution was more uniform.

The XRD pattern of the extruded AZ91-0.4Ce-xCa alloys is shown in Figure 2. The standard XRD patterns of basic structure in AZ91-0.4Ce magnesium alloy with different Ca content are shown in Appendix A. It can be seen that the extruded alloys contained α-Mg, β-Mg_17_Al_12_, Al_4_Ce, Al_2_Ce, and Al_2_Ca phases. Mg-Al-Zn ternary phases can exist only when the Zn/Al ratio exceeds 1/3 [18]. The absence of Zn-containing phases is due to the low Zn content in the alloy, where the Zn/Al ratio is about 9:1. The backscattered electron morphology of the extruded AZ91-0.4Ce-xCa alloys is shown in Figure 3. Table 2 shows the EDS spectrum results (Appendix A) of the points marked in Figure 3. As can be seen from Figure 3, the second phase of the mesh was broken into chunks after extrusion and distributed in strings along the extrusion direction. With the increase in Ca addition, the size of the second phase was gradually refined and the AZ91-0.4Ce-0.8Ca alloy exhibited the smallest second phase. The coarsening of the second phase size occurred when the Ca content increased to 1.2 wt.%.

Figure 4 shows the inverse pole figure (IPF) and corresponding pole figure of the extruded AZ91-0.4Ce-xCa alloys. It can be seen that equiaxed grains appeared regardless of adding Ca or not, indicating that dynamic recrystallization occurred during extrusion. As shown in Figure 4a, the grain size was the coarsest in the AZ91-0.4Ce alloy, with an average grain size of 12.6 μm, and the grain size was significantly refined with the addition of Ca. When the Ca content was 0.8 wt.%, the grain size was the smallest, with an average grain size of 4.7 μm. In addition, it can be seen that after the severe deformation of the alloy, the grains with different orientations are coordinated after deformation, and the texture is formed. The {0001} base texture during the plastic deformation of the alloy can be clearly seen. Specifically, the value of maximum pole intensity was 4.31 in the AZ91-0.4Ce-0.8Ca alloy, which indicates that AZ91-0.4Ce-0.8Ca has the weakest texture and the base texture has less impact on the mechanical properties.

### 3.2. Mechanical Properties of Extruded AZ91-0.4Ce-xCa Alloys

The mechanical properties of the extruded AZ91-0.4Ce-xCa alloys at room temperature are shown in Figure 5. The YS, UTS, and EL of the alloys increased first and then decreased when adding more Ca content. The AZ91-0.4Ce-0.8Ca alloy exhibited the best comprehensive properties; the YS, UTS, and EL were 238.7 MPa, 338.3 MPa, and 10.8%, respectively.

## 4. Discussion

### 4.1. Microstructure

The spherical Al_2_Ca phase was easily formed in Mg alloys containing Ca, which could effectively inhibit the forming of β-Mg_17_Al_12_ phases [19]. Majhi J et al. [20] pointed out that the electronegativity values of Mg, Al, Ce, and Ca were 1.31, 1.61, 1.12, and 1.00, respectively. The electronegativity differences between Ca and Al (0.61) and Ce and Al (0.49) were higher than that between Mg and Al (0.30). Therefore, adding a small amount of Ca and Ce to AZ91 alloys would preferentially precipitate the Al_2_Ca and Al_4_Ce phases. However, the addition of excessive Ca would coarsen the Al_2_Ca phase, as presented in Figure 1(d1). The second-phase particles in the extruded alloys were quantitatively counted, and the change trend of the second-phase volume fraction and average size are shown in Table 3.

The dispersed second-phase particles play an important role in grain boundary movement. The results show that the influence of second-phase particles on the grain growth rate is related to the radius of the second-phase particle (r) and the volume fraction per unit area (φ) [21]. At equilibrium, the stable grain size d has the following relationship with r and φ:d = 4r/3φ(1)

According to Equation (1), we calculate that the grain size of AZ91-0.4Ce-0.8Ca is the smallest. The size and distribution of the second phase and their effects on the grain size are shown in Figure 6. It can be seen that the smaller the size of the second phase, the higher the volume fraction per unit volume, the stronger the ability to hinder grain growth, and the finer the particles will be.

The network or long-strip second phase in the as-cast alloy was extruded and broken into a small granular distribution. Without the Ca added, the β-Mg_17_Al_12_ phase was distributed in bulk and the volume fraction of the second phase was 10.67%. Due to its body-centered cubic crystal structure, the β-Mg_17_Al_12_ phase is not in harmony with the close-packed hexagonal structure of the Mg matrix, which leads to the strong brittleness of the β-Mg_17_Al_12_ phase distributed along the grain boundaries, and the alloy is prone to stress concentration during deformation [22]. The addition of Ca inhibits the formation of the β-Mg_17_Al_12_ phase and has a positive effect on the plastic deformation behavior of the alloy at room temperature.

When the Ca content was 0.8 wt.%, the average size of the second phase was the smallest, and the distribution was relatively dispersed, which had the most positive second-phase strengthening effect on preventing the dislocation movement and improving the deformation resistance of the alloy. Moreover, the addition of Ca can refine the microstructure of the alloy, which is reflected in the thinning of the as-cast structure, the improvement in the recrystallization nucleation rate, and the nailing effect on the grain boundary. The fine dispersed Al_2_Ca phase can produce a typical particle-stimulated nucleation (PSN) effect on the thermally deformed structure, and play an important role in inducing DRX. On the other hand, the increased number of fine second phases has a nailing effect on the grain boundary, which inhibits the growth of recrystallized grains and refines the grains. The smaller the average grain size, the more internal grain boundaries there are, which could effectively hinder the dislocation movement and the coordination of plastic deformation of adjacent grains, so as to achieve the purpose of simultaneously enhancing the strength and plasticity of the alloy [23,24]. In the AZ91-0.4Ce-1.2Ca alloy, Al_2_Ca became coarse, and the stress concentration occurred during tensile deformation, which promoted the initiation and expansion of micro-cracks. In addition, excessive addition of Ca would increase the latent heat of solidification, increase the front temperature of the solid/liquid interface, and reduce the degree of supercooling and the nucleation rate, resulting in coarse grains, which would be unfavorable to the mechanical properties of the alloy.

To further identify the types of phases in the extruded alloys, TEM analyses were performed with an as-extruded AZ91-0.4Ce-0.8Ca alloy. Bright-field TEM micrographs are presented in Figure 7a–c. Corresponding diffraction patterns acquired from the selected regions marked by the arrows A, B in Figure 7a,b are shown in Figure 7e,f, respectively. The results are slightly different from the XRD results shown in Figure 2. We only found β-Mg_17_Al_12_ precipitates and Al_2_Ca phases in the alloy, but no Al_4_Ce phase. Among them, block-shaped β-Mg_17_Al_12_ precipitates with sizes of 400–500 nm and Al_2_Ca phases with sizes of about 200 nm were distributed as a round sphere in the matrix.

It is well known that the β-Mg_17_Al_12_ precipitates in the AZ91 alloy are a brittle phase, which is harmful to the mechanical properties of the alloy [25]. However, the addition of Ca would inhibit the precipitation of β-Mg_17_Al_12_ because the more stable Al_2_Ca phase would consume part of Al during its formation. Figure 7c shows a typical spherical phase of Al_2_Ca particles in the α-Mg matrix. The fine second phase can produce the typical PSN effect and can also nail the grain boundary to inhibit the grain growth, which is conducive to the improvement in the mechanical properties of the alloy [12]. As shown in Figure 7g, a large number of dislocations are clustered at the interface. Al_2_Ca can block the movement of the dislocation, producing a typical second-phase strengthening effect. When the content of Ca is 0.8%, the Al_2_Ca phase is fine and dispersive. As a result, more dislocation motion will be obstructed and dislocation stuffing will occur. According to the formula
τ = n·τ_0_,(2)
the stress concentration τ depends on the number of dislocations n [26]. The greater the n, the greater the stress concentration τ, and the more obvious the strengthening effect on the alloy.

### 4.2. Recrystallization and Texture

Recrystallization and texture are important factors affecting mechanical properties [27]. The deformation of magnesium alloys is easy to produce twins, and the continued deformation of twins usually requires symmetry conditions to make the crystal surface rotate in a certain direction, which usually makes the base surface tend to be parallel to the force direction [28,29,30]. This is the reason for the formation of basal texture, and strong basal texture has a negative effect on the mechanical properties of the alloy. As shown in Figure 4c, the grain (01−10) plane of the AZ91-0.4Ce-0.8Ca alloy was observed to be aligned with the deformation direction, resulting in the lowest Schmid factor for the {0001}<11−20> base plane slip when stress was applied along the extrusion direction. This suggests that the coordinated deformation ability of the alloy in response to external forces is stronger during plastic deformation, leading to high plasticity and low yield strength, in accordance with its mechanical properties [31]. Additionally, as the Ca content increases, the maximum texture density of the alloy is found to be 6.6, 4.6, 4.3, and 7.5. This indicates that the addition of Ca significantly weakens the texture, while excessive addition can result in texture enhancing.

When the Ca content is 0.8 wt.%, a high degree of dynamic recrystallization will produce fine grains, thus weakening the texture of the base surface and improving the mechanical properties [32]. The Statistical of the recrystallization grains and the Schmid factor of extruded AZ91-0.4Ce-xCa alloys are shown in Figure 8. After extrusion, the alloy deforms and produces twins, and the second phase is refined and dispersed. Therefore, DRX is believed to be generated by deformation twin-induced nucleation [33] and particle-stimulated nucleation (PSN) [34,35]. As shown in Figure 8c, the highest recrystallization degree (91.7%) was observed at a Ca content of 0.8 wt.%. In the extruded AZ91-0.4Ce-0.8Ca alloy, the fine and dispersed second-phase particles act as heterogeneous nucleation points for recrystallization grains with random orientation, promoting dynamic recrystallization behavior without obstructing dislocation motion [36,37]. However, at Ca contents of 0.4 and 1.2 wt.%, the degree of recrystallization is low due to the strong pinning effect of dislocation motion caused by coarse second-phase particles. The three alloys with higher sub-crystalline contents (Figure 8a,b,d) may have larger second-phase particles that exert drag force on sub-crystals [38], hindering their growth and affecting dynamic recrystallization nucleation, resulting in a decrease in dynamic recrystallization grains and an increase in sub-crystals.

## 5. Conclusions

The effects of Ca content on the microstructure and mechanical properties of AZ91-0.4Ce-xCa (x = 0, 0.4, 0.8, 1.2 wt.%) alloys were studied. The main conclusions are as follows:The phase composition of AZ91-0.4Ce-xCa alloys was α-Mg, β-Mg_17_Al_12_, Al_4_Ce, Al_2_Ce, and Al_2_Ca; Ca mainly existed in the form of Al_2_Ca. The addition of Ca can refine the second phase and grain size. AZ91-0.4Ce-0.8Ca exhibited the best refining effect. The newly formed Al_2_Ca phase was stable and could inhibit the formation of the β-Mg_17_Al_12_ phase.DRX behaviors happened after extrusion. When the Ca content was 0.8 wt.%, the average size of the second-phase particles was the smallest and the distribution was relatively dispersed, and the recrystallization degree was the largest (91.7%). The average grain size of AZ91-0.4Ce-0.8Ca was the smallest, at 4.75 μm.Grain-refinement strengthening and second-phase strengthening play an important role in improving the mechanical properties of the alloy.

## Figures and Tables

**Figure 1 materials-17-03359-f001:**
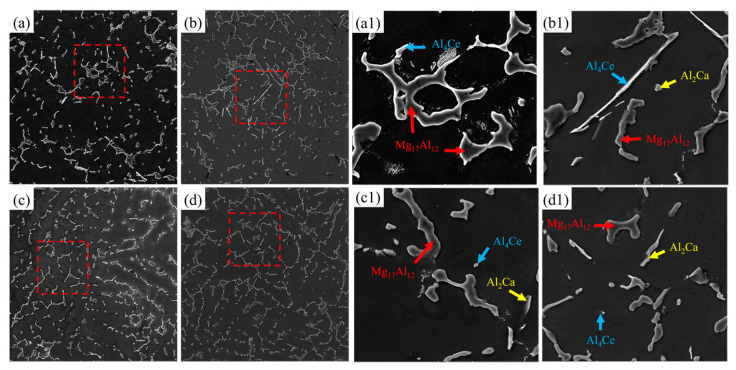
Microstructure scanning and energy spectrum analysis of as-cast AZ91-0.4Ce-xCa (x = 0, 0.4, 0.8, 1.2 wt.%) alloys. (**a**) AZ91-0.4Ce alloy, (**b**) AZ91-0.4Ce-0.4Ca alloy, (**c**) AZ91-0.4Ce-0.8Ca alloy, and (**d**) AZ91-0.4Ce-1.2Ca alloy. (**a1**–**d1**) correspond to selected parts in box (**a**–**d**), respectively.

**Figure 2 materials-17-03359-f002:**
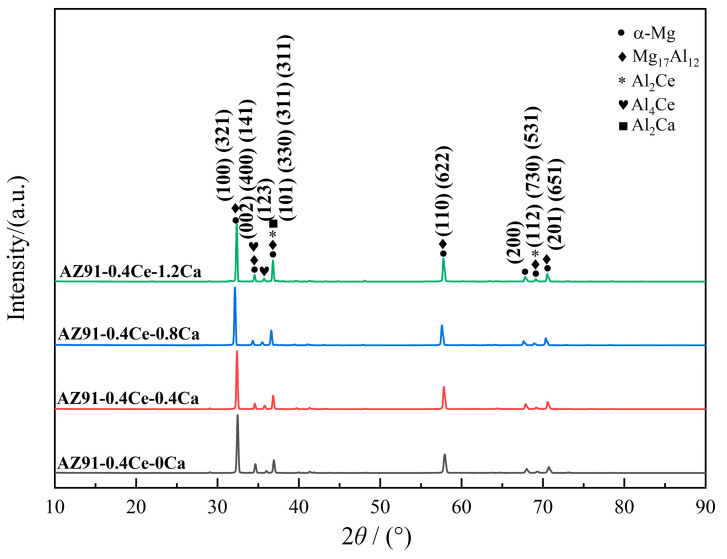
XRD pattern of extruded AZ91-0.4Ce-xCa (x = 0, 0.4, 0.8, 1.2 wt.%) alloys.

**Figure 3 materials-17-03359-f003:**
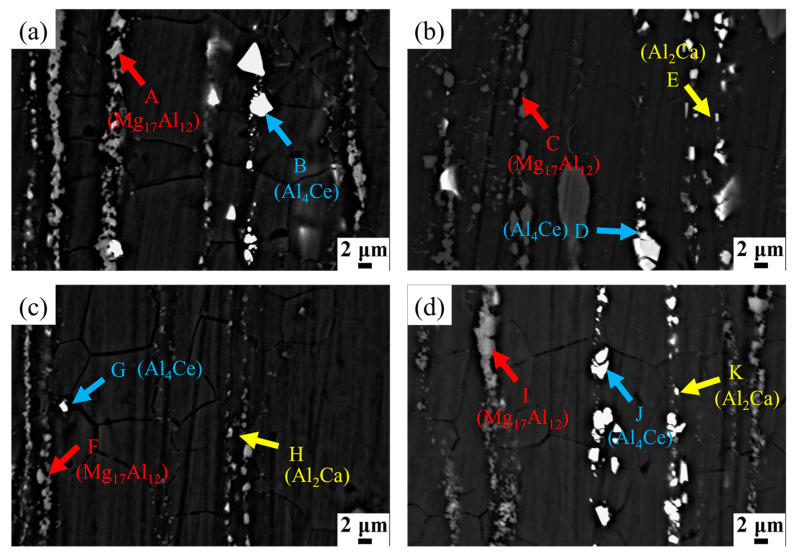
Microstructure scanning and energy spectrum analysis of extruded AZ91-0.4Ce-xCa (x = 0, 0.4, 0.8, 1.2 wt.%) alloys. (**a**) AZ91-0.4Ce alloy, (**b**) AZ91-0.4Ce-0.4Ca alloy, (**c**) AZ91-0.4Ce-0.8Ca alloy, and (**d**) AZ91-0.4Ce-1.2Ca alloy.

**Figure 4 materials-17-03359-f004:**
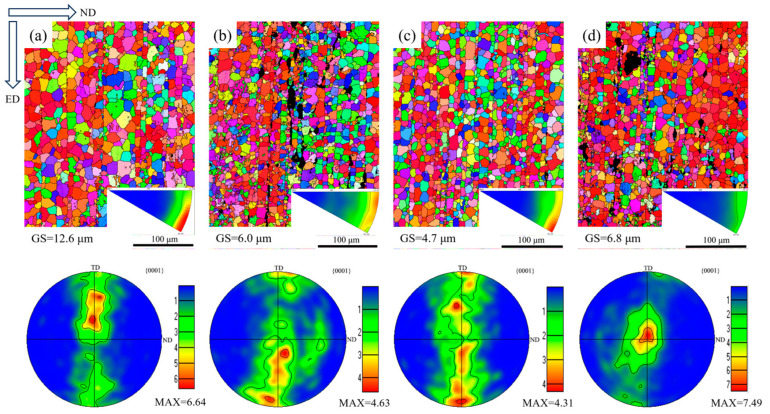
Extruded AZ91-0.4Ce-xCa (x = 0, 0.4, 0.8, 1.2 wt.%) alloys’ pole figure and inverse pole figure. (**a**) AZ91-0.4Ce alloy, (**b**) AZ91-0.4Ce-0.4Ca alloy, (**c**) AZ91-0.4Ce-0.8Ca alloy, and (**d**) AZ91-0.4Ce-1.2Ca alloy.

**Figure 5 materials-17-03359-f005:**
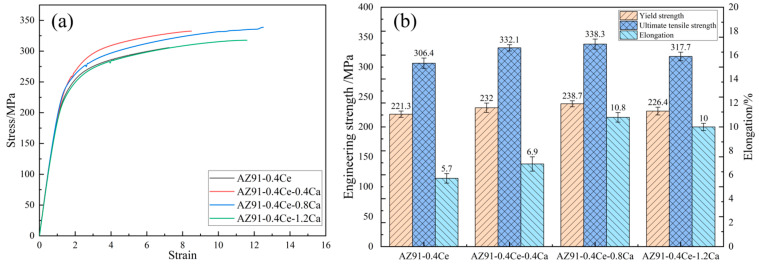
Mechanical properties of extruded AZ91-0.4Ce-xCa (x = 0, 0.4, 0.8, 1.2 wt.%) alloys at room temperature: (**a**) stress–strain curve; (**b**) variation trend of UTS, YS, and EL.

**Figure 6 materials-17-03359-f006:**
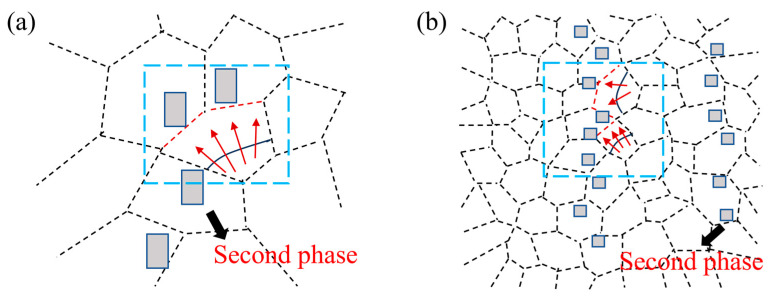
Effect of second-phase distribution and size on grain growth. (**a**) Coarse second phase; (**b**) fine dispersion of second phase (red arrows represent direction of grain boundary movement).

**Figure 7 materials-17-03359-f007:**
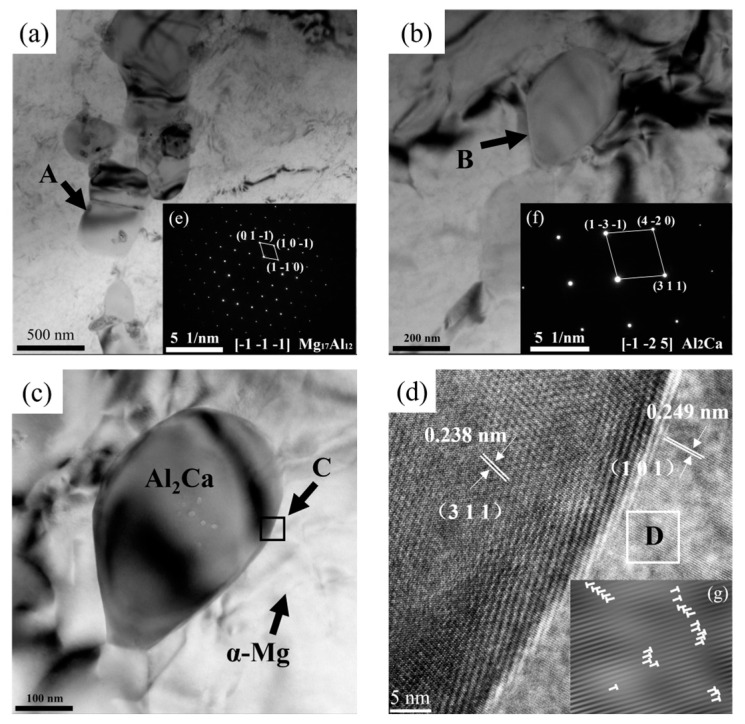
(**a**–**c**) Bright-field transmission electron microscope (TEM) micrographs of the as-extruded AZ91-0.4Ce-0.8Ca alloy; (**e**,**f**) the corresponding diffraction patterns of the selected particles taken from (**a**,**b**), indicated by the arrows A, B, respectively; (**d**) a high-resolution TEM (HRTEM) image of the zone C in (**c**); (**g**) an inverse fast Fourier transform (IFFT) image of the zone D in (**d**) (remark: the “T”-shaped symbols represent the dislocations).

**Figure 8 materials-17-03359-f008:**
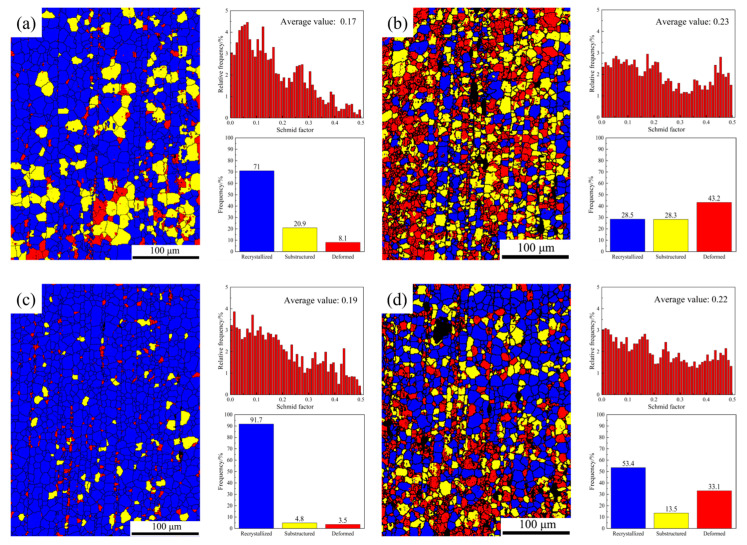
The Statistical of the recrystallization grains and the Schmid factor of extruded AZ91-0.4Ce-xCa (x = 0, 0.4, 0.8, 1.2 wt.%) alloys. (**a**) AZ91-0.4Ce alloy, (**b**) AZ91-0.4Ce-0.4Ca alloy, (**c**) AZ91-0.4Ce-0.8Ca alloy, and (**d**) AZ91-0.4Ce-1.2Ca alloy.

**Table 1 materials-17-03359-t001:** Chemical composition of cast alloys.

Sample	Component/(wt.%)				
	Mg	Al	Zn	Ca	Ce
AZ91-0.4Ce	Bal	9.12 ± 0.20	1.19 ± 0.06	0.00 ± 0.00	0.37 ± 0.10
AZ91-0.4Ce-0.4Ca	Bal	9.14 ± 0.25	1.22 ± 0.08	0.43 ± 0.07	0.37 ± 0.08
AZ91-0.4Ce-0.8Ca	Bal	9.33 ± 0.13	1.24 ± 0.11	0.81 ± 0.12	0.38 ± 0.05
AZ91-0.4Ce-1.2Ca	Bal	9.35 ± 0.36	1.12 ± 0.09	1.15 ± 0.15	0.38 ± 0.11

**Table 2 materials-17-03359-t002:** EDS elemental analysis of the phases marked in Figure 3.

Point	Composition/(at.%)					Point	Composition/(at.%)				
	Mg	Al	Zn	Ce	Ca		Mg	Al	Zn	Ce	Ca
A	59.12	39.62	1.26	--	--	G	3.25	74.79	1.02	20.94	--
B	3.65	80.21	--	16.14	--	H	8.95	63.25	--	--	27.80
C	60.85	37.65	1.50	--	--	I	60.49	38.36	1.15	--	--
D	5.28	76.10	--	18.62	--	J	2.31	74.49	--	23.20	--
E	6.32	67.31	--	--	26.37	K	4.37	63.48	2.64	--	29.51
F	58.21	39.42	2.37	--	--						

**Table 3 materials-17-03359-t003:** Volume fraction and average size of second phase of extruded AZ91-0.4Ce-xCa (x = 0, 0.4, 0.8, 1.2 wt.%) alloys.

	AZ91-0.4Ce	AZ91-0.4Ce-0.4Ca	AZ91-0.4Ce-0.8Ca	AZ91-0.4Ce-1.2Ca
Volume fraction/%	10.67	7.09	8.11	9.28
Average size/μm	5.14	3.98	2.41	3.95

## Data Availability

The data that support the findings of this study are available from the corresponding author upon reasonable request.

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
