# Peer review of "Modification of Microstructure and Mechanical Properties of Extruded AZ91-0.4Ce Magnesium Alloy through Addition of Ca"

_materials, 2024, doi:10.3390/ma17133359_

Round 1

Reviewer 1 Report

Comments and Suggestions for Authors

The paper describes experimental results on the synthesis and characterization of Mg alloys. The results are clear and well presented. As minor comment, I suggest to move the results reported in the materials and methods (e.g. SEM, chemical composition) in the results section.

Reviewer 2 Report

Comments and Suggestions for Authors

Extensive editing of the English language is required. Many sentences cannot be easily understood. In the attached file, some of them are marked in the commentaries or marked in yellow.

In page 2, line 53, change "intermediate" to "master". It's much more usual.

In Fig.1, Fig.3, the phase identifications in red cannot be read. Please change size/colour of phase identification.

In Fig.2, the phase symbols are only incorporated in AZ91-0.4Ce-1.2Ca. Modify the figure.

In Fig.5 b), please incorporate the deviation values for every measured property and alloy.

In Fig.7, it must be increased the size of the figures. Not readable.

In Fig.8, it must be increased the size of the figures. Not readable.

Comments on the Quality of English Language

Extensive editing of the English language is required. Many sentences cannot be easily understood. In the attached file, some of them are marked in the commentaries or marked in yellow. Verb tenses are mixed in the present and past tense.

In page 2, line 53, change "intermediate" to "master". It's much more usual.

In Fig.1, Fig.3, the phase identifications in red cannot be read. Please change size/colour of phase identification.

In Fig.2, the phase symbols are only incorporated in AZ91-0.4Ce-1.2Ca. Modify the figure.

In Fig.5 b), please incorporate the deviation values for every measured property and alloy.

In Fig.7, it must be increased the size of the figures. Not readable.

In Fig.8, it must be increased the size of the figures. Not readable.

Reviewer 3 Report

Comments and Suggestions for Authors

The authors of this manuscript present a study regarding the effect of Ca addition on microstructure, mechanical properties of extruded AZ91-0.4Ce-xCa (x=0, 0.4, 0.8, 1.2wt.%) alloy.

However several details should be addressed before publication:

 1. It seems that the novelty and Aims of manuscript is not clear indicated. Authors should improve that.

2. Error bar of wt % in Table 1 should be inserted.

3. EDX mapping of surface in Fig 3 and 7 should be provided to clarify the results

4. How was prepared the samples for TEM / SEM investigation?

5. (hkl) plane should be added in Fig.2

Comments on the Quality of English Language

 Moderate editing of English language required
